# Contamination of Arctic Lakes with Persistent Toxic PAH Substances in the NW Part of Wedel Jarlsberg Land (Bellsund, Svalbard)

**Sara Lehmann-Konera [1,*], Marek Ruman [2], Łukasz Franczak [1] and Żaneta Polkowska [3]**

[1] Institute of Earth and Environmental Sciences, Faculty of Earth Sciences and Spatial Management, Maria Curie-Skłodowska University in Lublin, Kraśnicka 2d Ave., 20-718 Lublin, Poland; lukasz.franczak@poczta.umcs.lublin.pl

[2] Institute of Earth Sciences, Faculty of Natural Sciences, University of Silesia, Będzińska 60 St., 41-200 Sosnowiec, Poland; marek.ruman@us.edu.pl

[3] Department of Analytical Chemistry, Faculty of Chemistry, Gdańsk University of Technology, 11/12 Narutowicza St., 80-233 Gdańsk, Poland; zanpolko@pg.edu.pl

[*] Correspondence: s.lehmann-konera@poczta.umcs.lublin.pl; Tel.: +48-81-537-6873

**Abstract:** The expansion of glacier-free areas in polar regions favours the appearance of lakes in the non-glaciated parts of glacier basins. This paper presents the differentiation of organic compound concentrations in fifty-four Arctic lakes collected in four locations (Logne Valley, in the vicinity of the Scott, Renard and Antonia glaciers). We cover meteorological measurements, chemical analysis of sixteen dioxin-like compounds (Polycyclic Aromatic Hydrocarbons (PAHs)), formaldehyde (HCHO), sum parameters of phenolic compounds (∑phenols) and dissolved organic carbon (DOC). The most contaminated with PAH compounds were lakes exposed to the influence of the Greenland Sea (Logne Valley lakes) and to the prevailing winds (Scott and Renard lakes). Interpretation of the PAH compounds results allowed for identification of pyrogenic sources as the main sources of PAH compounds in the year 2012. The highest levels of HCHO and ∑phenols were observed for the Scott lakes, while the highest DOC levels were noted in Antonia lakes.

**Keywords:** Arctic; lakes; polycyclic aromatic hydrocarbons; organic compounds; cross-border pollution

## 1. Introduction

Global atmospheric circulation and individual properties of the natural environment of the Arctic make it a recipient of pollutants emitted in urbanised and industrialised areas at lower latitudes (North America, Eurasia) [1,2]. A report by experts of the Arctic Monitoring Assessment Program (AMAP) draws particular attention to the problem of Long-Range Transport of Atmospheric Pollutants (LRTAP) from Eurasia and North America to the Arctic. Moreover, they emphasise the strong need to supplement data on concentrations of, among others, Polycyclic Aromatic Hydrocarbons (PAHs) in the surface waters of the Arctic [3]. Due to the intensification of human activity in Eurasia, increasing quantities of harmful chemical compounds reach the Arctic as a result of LRTAP [1,2]. Thus far, the presence of such pollutants as heavy metals (e.g., Pb and Hg), radioactive isotopes and numerous organic compounds (e.g., pesticides, PAHs, polychlorinated biphenyls (PCBs)) has been detected in the Arctic. Pollutants reaching the Arctic can originate both from natural sources, including forest fires and volcanic eruptions [4], and from anthropogenic sources. Industry in Asia is estimated to account for more than 50% of global emissions of PAHs to

the atmosphere [3]. The Svalbard Archipelago is particularly susceptible to the accumulation of a broad range of chemical compounds considered as pollutants. Both dry and wet deposition of pollutants is favoured by: the climatic conditions of Polar regions (low temperatures reduce the volatility of Persistent Organic Pollutants (POPs), including PCBs and PAHs); geographic location (due to its location in the gap between land massifs, both air masses and marine currents are a medium for pollutants); and orographic factors (the mountainous land relief prevailing on the islands is a natural barrier to air masses inflowing from the south) [5,6].

The increases in average air temperature recorded in the 20th century in the High Arctic have been higher than those for the hemisphere as a whole [7]. Meanwhile, in the 21st century, the Arctic is warming twice as fast as any other place on earth [8,9]. This favours the acceleration of intensive melting of glaciers and their rapid retreat, leading to an expansion of the ice-free area [9–11] with an abundance of lakes and ponds. Arctic freshwater ecosystems, such as lakes and ponds, are extremely sensitive to climate changes [12,13], local pollution inputs and the deposition of contaminants from LRTAP [14,15], and even increases in ultraviolet (UV) radiation [13,16–18]. High Arctic lakes are considered to be relatively productive ecosystems in a very demanding polar environment. These small oases of life at higher latitudes are one of the least studied water ecosystems [13]. Physicochemical characteristics, main ions, trace elements and organic carbon in Canadian Arctic lakes and ponds are very well described [16–19]. Research conducted in Norwegian and Russian Arctic lakes provides information about levels of trace elements [20], main ions and organic carbon [21] in water and polychlorinated biphenyls (PCBs) in sediments [22]. There are several works focused on analysis of the inorganic chemistry of Svalbard lakes' waters [23–27] and sediments [28]. There are numerous publications giving an overview of concentrations of polycyclic aromatic hydrocarbons (PAHs) in lake sediments of Svalbard [29–32], but only a few about PCBs [32]. Although knowledge about the occurrence of PAHs in Svalbard freshwater has increased in recent years [4,33–35] only two works refer to lake water [34,35].

The short review above indicates the need to expand the knowledge about the occurrence and levels of persistent organic compounds, such as PAHs, in the waters of Svalbard lakes. In comparison to lake sediments, the work in this area is scarce, and the number of lakes discussed in the literature is limited only to one, which is located in the vicinity of the Hornsund Polar Station. This paper aims to compare the chemical composition of organic compounds in four groups of Arctic lakes by providing insights into the differentiation of concentration of PAHs, Σphenols, formaldehyde (HCHO) and dissolved organic carbon (DOC) in 54 lakes located in the NW part of Wedel Jarlsberg Land. Particular attention will be given to PAH compounds due to their confirmed toxic effect on living organisms. In this work, the authors also attempted to indicate the likely sources of emission of PAHs that have been detected by use of diagnostic ratios.

## 2. Materials and Methods

### 2.1. Samples and Sampling Location

Lake water samples were collected in the Logne Valley (77°32′ N, 13°59′ E; N = 5), in intra-marginal zones of the Scott Glacier (77°33′ N, 14°25′ E; N = 19) and the Renard Glacier (77°32′N, 14°29′ E; N = 27), and in the extra-marginal zone of the Antonia Glacier (77°32′ N, 14°53′ E; N = 3), all in the NW part of Wedel Jarlsberg Land (Figure 1) from 25 July to 8 August 2012. Fifty-four surface water samples were collected manually to airtight bottles (1 litre volume) triple-rinsed with the sample. To avoid the loss of analytes to headspace, bottles were filled without air bubbles. All samples were stored and transported to the laboratory in dark bottles at a temperature of approximately 4 °C to ensure unchanged chemical composition. Moreover, to mitigate the impact of the sampling containers, a blank sample was used as a control.

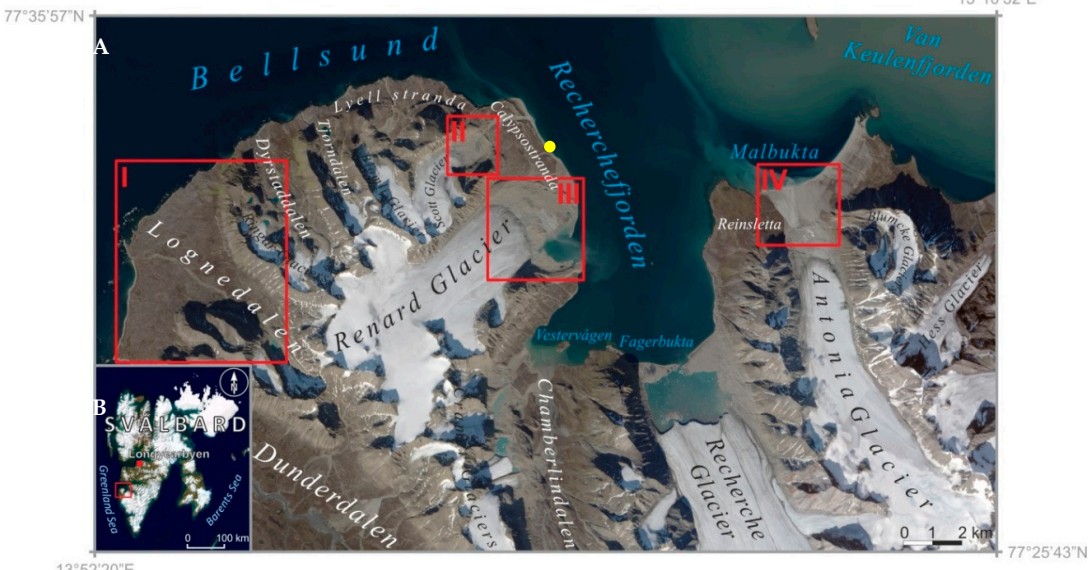

**Figure 1.** Location of the study area: I—Logne Valley; 2—intra-marginal zone of the Scott Glacier; III—intramarginal zone of Renard Glacier; IV—extra-marginal zone of Antonia Glacier and location of automatic weather station (yellow dot). Source of aerial photo: **A** [36], **B** [37].

The study area is delimited to the north by the Bellsund Fjord, which towards the south becomes the Recherche Fjord. In this part of the area, lakes occur in the intra-marginal zones of the Scott and Renard glacier forefields, which are the most distant from the influence the sea. Towards the east, Bellsund becomes the Van Keulen Fjord, where lakes are situated in the extra-marginal zone of the Antonia Glacier. To the west, the study area is surrounded by the open Greenland Sea, which influences, among others, the lakes within the wide Logne Valley. Its degree of glaciation as compared to the other investigated areas is small (approximately 10%), and the small glaciers cover a total area of 2.2 km² and occur only in its upper part at 250–500 m a.s.l. The largest glaciers related to the occurrence of the investigated lakes are Renard and Antonia. The former has a surface area of 29 km², a length of 10 km and a width ranging from 2.5 km in the ablation zone to over 6 km in the accumulation zone. Its front reaches as low as 50 m a.s.l. and its highest part reaches 750 m a.s.l. The Antonia Glacier has an area of 29 km², a length of 11.5 km and a width of 1.5–4.5 km. Its hypsometry spans from 100 to 800 m a.s.l. Markedly smaller is the Scott Glacier, occupying an area of 4.5 km². Its length is 3.5 km, and its width spans 1–1.5 km. The glacier front reaches approximately 90 m a.s.l., and its highest parts reach 600 m a.s.l. [36]. Nowadays, the glaciers occurring in the area are experiencing a deep recession and their forefields display marked morphogenetic dynamics [38]. Most lakes were formed in the intra-marginal zones of glacier forefields. They were mostly formed on moraine sediment, which consists of lithologically diversified coarse-grain sediment (mainly boulders, rubble and gravel) with an admixture of sand and silt, within sandur areas of better-rounded and better-sorted rock material. Due to the influence of various exogenous processes connected with the activity of glaciers, and the further landscape changes that have been taking place since their disappearance, the lake basins have been formed as a result of glacial and fluvio-glacial erosion and accumulation, dead-ice melt and permafrost degradation [39,40].

### 2.2. Chemical Analysis

Immediately after delivering the samples to the laboratories, quantitative analyses of organic compounds were conducted. For the analyses of PAH compounds, Gas Chromatography coupled with Mass Spectrometry Technique was used. The analytical procedure used for determination of PAHs is described in detail by Kozak et al. [4]. Formaldehyde (HCHO) and sum of phenols (Σphenols) were determined by Spectrophotometry Method, while for dissolved organic carbon

(DOC) the method of catalytic combustion (oxidation) was employed, using an NDIR detector (Table 1). All blanks were prepared with Milli-Q de-ionised water.

**Table 1.** Validation parameters and technical specifications used in analytical procedures.

| Parameters (Acronyms) | Measurement Range | LOD *** | LOQ *** | Measurement Instrumentation/Reagents |
|---|---|---|---|---|
| PAHs * | | | | |
| Naphthalene (NP) | 1.02–3500 | 0.034 | 1.02 | |
| Acenaphthylene (ACY) | 0.012–1000 | 0.004 | 0.012 | |
| Acenaphthene (ACE) | 0.012–1000 | 0.004 | 0.012 | |
| Fluorene (FL) | 0.005–1000 | 0.002 | 0.005 | Gas Chromatograph 7890A (Agilent Technologies, Santa Clara, CA, USA) coupled with a mass spectrometer (5975C inert MSD—Agilent Technologies), detector (Agilent Technologies 5975C) with electron ionisation (SIM mode) / Dichloromethane, Methanol Sigma-Aldrich Company; Naphthalene-d8, Benzo(a)anthracene-d12, Supelco; Mixtures of 16 PAHs (2000 μg/mL in dichloromethane), Restek Corporation |
| Phenanthrene (PHE) | 0.008–1000 | 0.003 | 0.008 | |
| Anthracene (ANT) | 0.023–1000 | 0.008 | 0.023 | |
| Fluoranthene (FLA) | 0.042–1000 | 0.014 | 0.042 | |
| Pyrene (PYR) | 0.084–1000 | 0.028 | 0.084 | |
| Chrysene (CHR) | 0.007–1000 | 0.002 | 0.007 | |
| Benzo(b)fluoranthene (BbF) | 0.042–1000 | 0.014 | 0.042 | |
| Benzo(k)fluoranthene (BkF) | 0.007–1000 | 0.002 | 0.007 | |
| Benzo(a)pyrene (BaP) | 0.017–1000 | 0.006 | 0.017 | |
| Benzo(a)anthracene (BaA) | 0.005–1000 | 0.002 | 0.005 | |
| Benzo[g,h,i]perylene (BghiP) | 0.004–1000 | 0.001 | 0.004 | |
| Indeno(1,2,3-cd)pyrene (IcdP) | 1.29–1000 | 0.431 | 1.29 | |
| Dibenz(a,h)anthracene (DahA) | 0.042–1000 | 0.014 | 0.042 | |
| Dissolved Organic Carbon (DOC) ** | 0.030–10.0 | 0.030 | 0.100 | Total Organic Carbon Analyser TOC-V$_{CSH/CSN}$, SHIMADZU, Japan, Potassium Hydrogen Phtalate standard |
| Formaldehyde (HCHO) ** | 0.02–1.5 | 0.005 | 0.015 | Spectrophotometer 6300, Jenway Absorbance measured at 585 nm |
| Sum of phenols (∑phenols) ** | 0.025–5.00 | 0.001 | 0.003 | Absorbance measured at 495 nm |

* ng/L, ** mg L$^{-1}$; *** Both the limit of detection (LOD) and the limit of quantitation (LOQ) were calculated based on the standard deviation of response (s) and the slope of the calibration curve (b) according to the formulas: LOD = 3.3(s/b), LOQ = 10(s/b).

Environmental samples are characterised by various matrix compositions and require that the analytical procedures applied in determining their individual components be validated against certified reference materials (CRM) concordant with ISO Guide 34:2009 and ISO/IEC 17025:2005. Certified reference materials (low-level PAHs in acetone (QC1223) and TOC in water matrix ((QC3308)), Sigma-Aldrich Chemie Gmbh, Munich, Germany) were used in the procedure-validation stage. The data obtained during the analyses were subject to strict quality control (QC) procedures in order to ensure high quality results. Values of PAH compounds and DOC in CRMs were within the confidence interval, with high recovery (85%–105%) and standard deviation (RSD) 4%–10%. The measurements of HCHO were made in accordance with norms ISO 8466-1, and Σphenols according to DIN 38,402 A51. Demineralised water type Mili-Q (Mili-Q®

Ultrapure Water Purification Systems, Millipore® production, Darmstadt, Germany was applied for the determination of various targets of analyte groups.

### 2.3. *Factors for Results Analysis*

#### 2.3.1. Meteorological Conditions

During a polar expedition from July 13 to August 24 of 2012, a portable weather station (Campbell Scientific, CR10X, Datalogger for Measurement and Control) was placed approximately 200 m from the seashore at an altitude of 23 m a.s.l. on the seaside terrace in the vicinity of the Calypsobyen Polar Station (Bellsund Fjord, Spitsbergen). The weather conditions, including changes in air temperature, occurrence of precipitation and wind directions were previously discussed in other works [41–44].

The authors assumed that the weather conditions measured in the Calypsostranda seaside terrace corresponded to the entire study area of Bellsund Fjord, and Figure 2A–C were prepared on this basis.

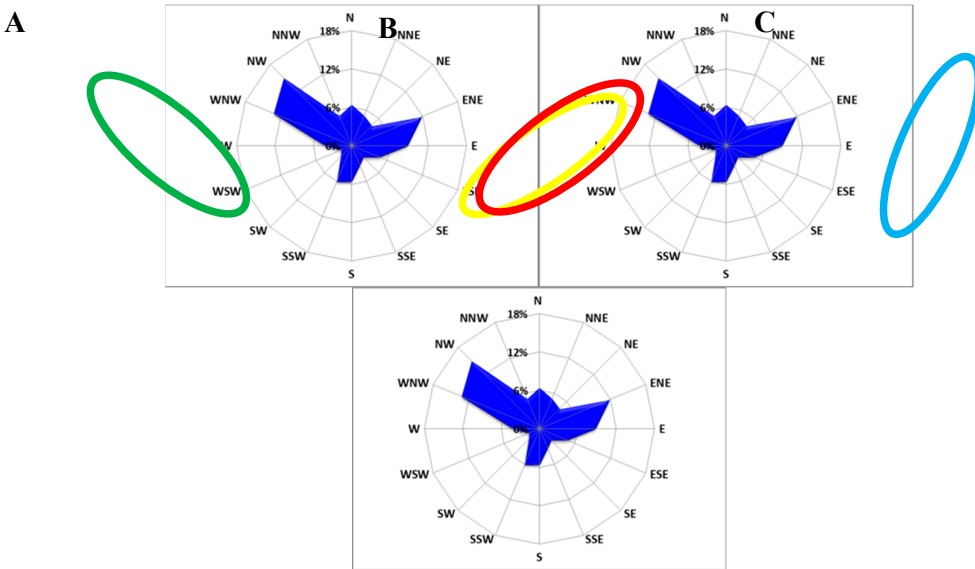

**Figure 2.** Percentage of prevailing wind directions on the Calypsostranda area (NW Wedel Jarlsberg Land) before and during the sampling campaign (13.07–08.08.2012). Circles represent the sampling site exposure to wind: **A**—Logne Valley; **B**—intra-marginal zones of the Scott and Renard glaciers; **C**—extra-marginal zone of Antonia Glacier.

As shown in Figure 2, between the 13th of July and the 8th of August, NW Wedel Jarlsberg Land was under the influence of winds coming from NW (15%), WNW (13%), ENE (12%) and E (9%). The percentage contribution of winds coming from N, NNE, ESE, S, SSW, NNW ranged between 5% and 6%. Winds coming from NE, SE, SSE, SW, WSW, W hardly ever have a greater influence on the studied area (only 2%–4%).

#### 2.3.2. Statistical Methods

In order to characterise the differentiated chemical composition of the lakes, the authors analysed the occurrence in water samples of: 16 PAH compounds; HCHO; Σphenols; and DOC. Principle component analysis (PCA) was computed with the software package STATISTICA 6.1 (StatSoft Inc., Tulsa, OK, USA).

2.3.3. PAH Indicator Ratio

Petrogenic and pyrogenic substances consist of the same compounds. Petrogenic PAH compounds are formed in relatively low temperatures, while pyrogenic in temperatures ranged from 350 to 1200 °C (low-oxygen or no-oxygen conditions). Most importantly, pyrogenic and pyrolytic substances originating from different sources actually have measurably different amounts of some PAHs. To distinguish these two sources of PAH compounds, their distributions or relative concentrations are often used. PAH indicator ratios are commonly used to trace the origin of these organic compounds in air, sediments and water alike. PAH indicator ratios make it possible to determine the potential sources of these compounds as being from petrogenic or pyrogenic processes, but are also helpful for more specifically defining the source of PAHs as being from: coal combustion (BaA/(BaA + CHR)); combustion of fossil fuel, grass, wood and coal (FLA/(FLA + PYR)); petroleum, grass, wood and coal combustion (IcdP/(IcdP + BghiP)); traffic and non-traffic emissions (BaP/BghiP); or petrol or diesel emissions (FL/(FL + PYR)) [4,29,33,45,46].

## 3. Results

### 3.1. General Overview of Organic Compound Concentrations in Lakes

This paper presents results of analyses of 16 PAH compounds (NP; ACY; ACE; FL; PHE; ANT; FLA; PYR; CHR; BbF; BkF; BaP; BaA; BghiP; IcdP; DahA), HCHO, $\Sigma$phenols and DOC concentrations for 54 Arctic lakes (Table 2) (Figure 2).

The analysis of the results reveals differences between chemical composition of organic compounds in the lakes of NW Wedel Jarlsberg Land (Table 2). The highest value of mean $\Sigma_{16}$PAH compounds was noted for Logne lakes (530 ng/L), with lower values for the Scott and Renard lakes (276 and 104 ng/L, respectively) and the lowest for Antonia lakes (47.8 ng/L) (Figure 3A,B). The composition of PAH compounds in Logne lakes is the most varied of all the studied group of lakes. Compounds such as NP, ACY, ACE, FL, PHE, ANT, FLA, PYR, CHR, BbF, BkF, BaA were detected in all lakes from Logne Valley. In the lakes from intra-marginal zones of the Scott and Renard glaciers, compounds such as NP, FL, PHE, FLA, PYR were detected in most of the samples, while ACY, ANT, CHR, BbF, BkF, BaA were mostly noted as below limit of detection. It is interesting that ACE was detected in 83% of samples collected from the Renard Glacier marginal zone and only in 16% of samples from the Scott Glacier marginal zone. In the lakes from extra-marginal zone of the Antonia Glacier only NP was detected in all water samples, while ACE, FLA and BaA were detected in two out of three samples. The rest of the PAH compounds in these lakes were below their LODs. The highest values (above 100 ng/L) were noted for NP and PHE in Logne (1,04 and 110 ng/L, respectively), Scott (236 and 357 ng/L, respectively) and Renard lakes (142 and 192 ng/L, respectively), and also for PYR in the Renard lakes (123 ng/L). Organic compounds such as BghiP, IcdP, DahA were below limit of detection in 96% of water samples from the Renard lakes and in all the Logne, Scott and Antonia lakes.

**Table 2.** Environmental datasets for all collected lakes.

| Determined Compounds | Logne (L) N = 5 | | | Scott (S) N = 19 | | | Renard (R) N = 27 | | | Antonia (A) N = 3 | | |
|---|---|---|---|---|---|---|---|---|---|---|---|---|
| | Min | Max | Median | Min | Max | Median | Min | Max | Median | Min | Max | Median |
| NP * | 77.0 | 1 037 | 211 | 14.3 | 236 | 56.9 | 2.68 | 142 | 28.0 | 40.8 | 51.3 | 46.9 |
| ACY * | 0.287 | 6.10 | 1.94 | <LOD | <LOD | <LOD | <LOD | 5.89 | <LOD | <LOD | <LOD | <LOD |
| ACE * | 3.08 | 16.5 | 3.57 | <LOD | 11.2 | <LOD | <LOD | 11.9 | 1.70 | <LOD | 0.247 | 0.129 |
| FL * | 5.23 | 19.9 | 5.58 | <LOD | 26.4 | 12.7 | <LOD | 23.6 | 3.62 | <LOD | <LOD | <LOD |
| PHE * | 30.4 | 110 | 49.8 | 34.8 | 357 | 85.0 | <LOD | 192 | 22.4 | <LOD | <LOD | <LOD |
| ANT * | 2.05 | 15.2 | 3.21 | <LOD | 4.45 | <LOD | <LOD | 93.2 | <LOD | <LOD | <LOD | <LOD |
| FLA * | 3.64 | 16.5 | 4.93 | 4.81 | 25.1 | 15.1 | <LOD | 73.0 | 1.76 | <LOD | 1.47 | 0.721 |
| PYR * | 3.58 | 13.5 | 4.87 | 24.7 | 60.3 | 44.5 | <LOD | 123 | 1.53 | <LOD | <LOD | <LOD |
| CHR * | 0.978 | 7.31 | 1.66 | <LOD | 55.3 | 4.06 | <LOD | 41.7 | <LOD | <LOD | <LOD | <LOD |
| BbF * | 0.564 | 9.71 | 1.90 | <LOD | 47.9 | <LOD | <LOD | 24.9 | <LOD | <LOD | <LOD | <LOD |
| BkF * | 0.564 | 9.71 | 1.90 | <LOD | 47.9 | <LOD | <LOD | 24.9 | <LOD | <LOD | <LOD | <LOD |
| BaP * | <LOD | 1.54 | <LOD | <LOD | 24.9 | <LOD | <LOD | 6.75 | <LOD | <LOD | <LOD | <LOD |
| BaA * | 0.934 | 5.80 | 1.62 | <LOD | 62.9 | 0.47 | <LOD | 9.52 | <LOD | <LOD | 1.29 | 0.317 |
| BghiP * | <LOD | <LOD | <LOD | <LOD | <LOD | <LOD | <LOD/ | 0.640 | <LOD | <LOD | <LOD | <LOD |
| IcdP * | <LOD | <LOD | <LOD | <LOD | <LOD | <LOD | <LOD | 0.776 | <LOD | <LOD | <LOD | <LOD |
| DahA * | <LOD | <LOD | <LOD | <LOD | <LOD | <LOD | <LOD | <LOD | <LOD | <LOD | <LOD | <LOD |
| $\Sigma_{16}$PAHs * | 151 | 1 250 | 335 | 115 | 487 | 267 | 19.9 | 295 | 76.3 | 42.2 | 54.1 | 47.0 |
| $\Sigma$phenols ** | 0.007 | 0.061 | 0.010 | 0.005 | 2.00 | 0.016 | 0.005 | 0.703 | 0.016 | 0.029 | 0.090 | 0.072 |
| HCHO ** | 0.01 | 0.02 | 0.01 | 0.01 | 0.92 | 0.01 | 0.01 | 0.02 | 0.01 | 0.04 | 0.10 | 0.10 |
| DOC ** | 1.66 | 2.42 | 2.13 | 0.134 | 104 | 0.529 | 0.347 | 1.88 | 0.798 | 1.45 | 19.2 | 6.27 |

* ng/L, ** mg/L.

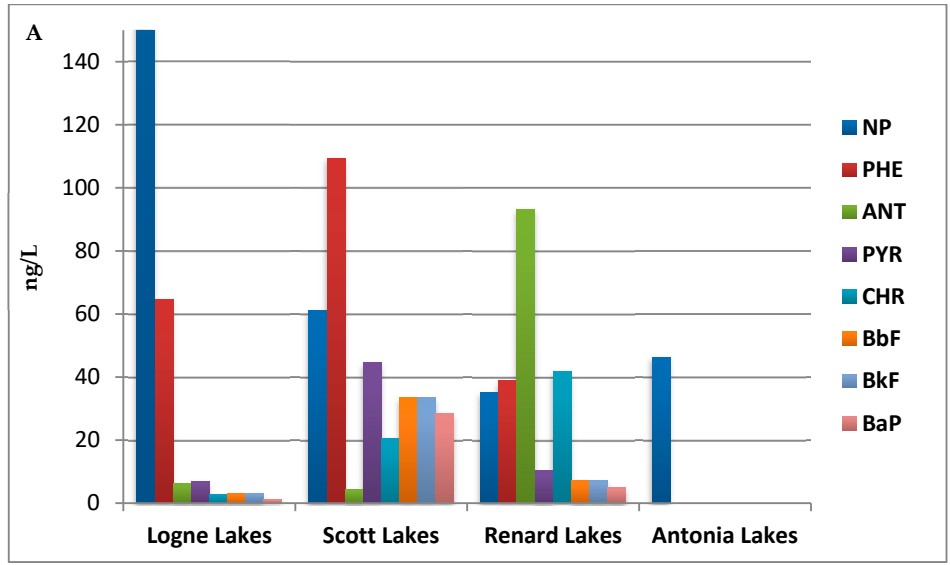

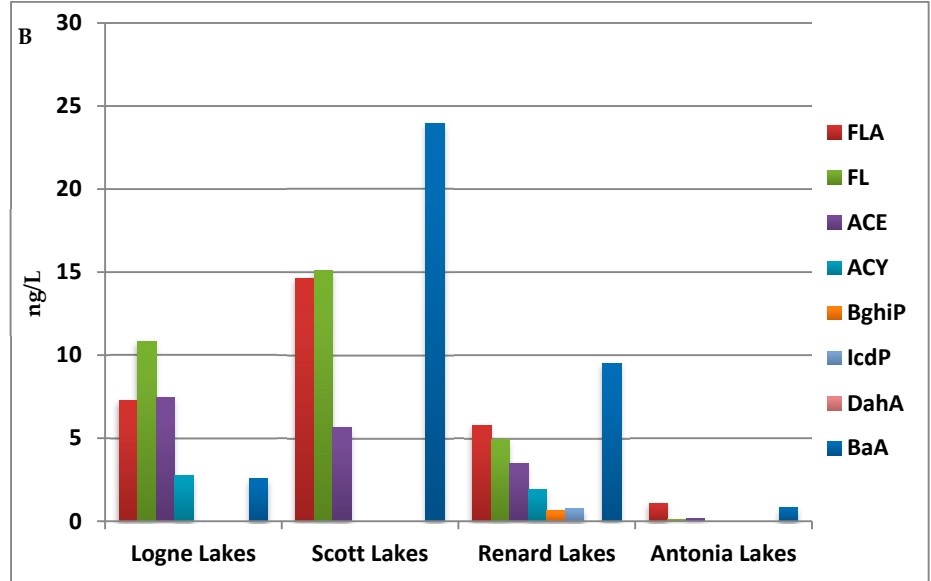

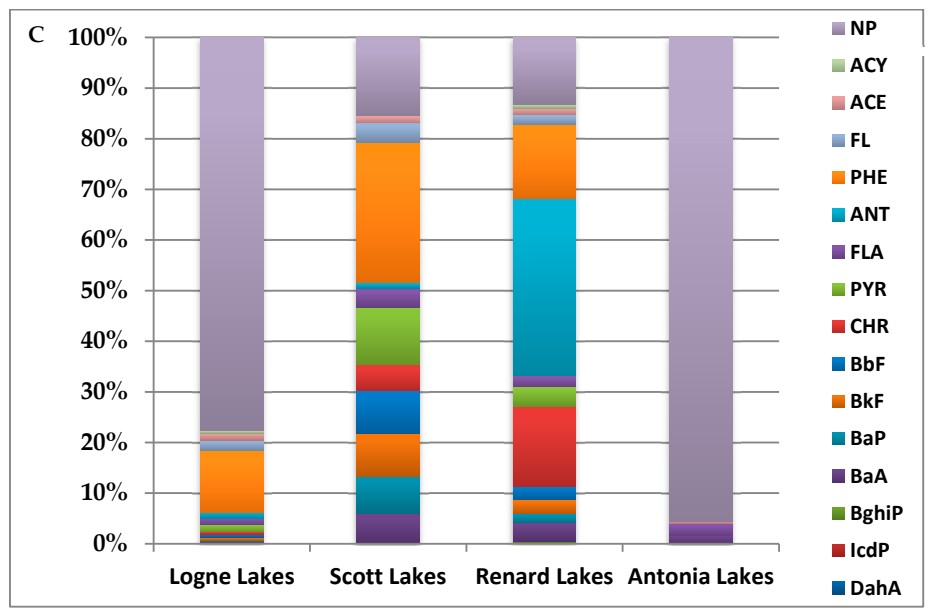

**Figure 3.** Distribution of detected Polycyclic Aromatic Hydrocarbons (PAH) compounds in studied water samples: (**A**), (**B**)—differences in mean concentration; (**C**)—percentage contribution in the mean value of the $\Sigma_{16}$PAHs.

The difference in chemical composition of PAH compounds is clearly visible in Figure 3C. The Logne, Scott and Renard lakes have varied composition of PAH compounds, while Antonia lakes present very low differentiation. Both in Logne and Antonia lakes, naphthalene (NP) has the highest percentage contribution (77.6% and 95.5%, respectively) and is also one of the leading PAH compounds in lakes from Scott and Renard. Another leading compound in the Logne lakes is phenantrene (PHE) (12.1%). The rest of the PAHs in the Logne lakes have similar percentage contributions to one another (in the range 0.208%–2.03%). The highest percentage contribution in the Scott lakes is for PHE (27.7%), NP (15.5%) and PYR (11.3%), while in the Renard lakes it is for ANT (35.1%), CHR (15.7%), PHE (14.7%) and NP (13.2%). The mean values of other determined organic compounds such as formaldehyde and phenols that were determined for the lakes also differ significantly. Meanwhile, water from the Logne lakes was the most polluted by PAH compounds and characterised by very low concentrations of $\Sigma$phenols as well as HCHO (0.020 mg/L and 0.01 mg/L, respectively) in comparison to the other groups of lakes. In water samples from the Renard lakes, mean values of phenolic compounds (0.053 mg/L) were higher than those of formaldehyde (0.01 mg/L), while in the Antonia lakes they were at similar levels of concentration to one another (0.064 mg/L and 0.08 mg/L). The highest mean values for $\Sigma$phenols and HCHO were detected in the Scott lakes (0.323 mg/L and 0.11 mg/L, respectively).

### 3.2. Statistical Analysis

Principal Components Analysis (PCA) has been applied to a set of 19 variables (Table 2). The data included HCHO, 16 compounds from the PAHs group, and the summative parameters DOC and $\Sigma$phenols. Based on standardised data, two principal components F1 and F2 were distinguished which together explained 45% of the initial data variability (30% and 15% of variance for F1 and F2, respectively) (Figure 4). The significance of these principal components was highlighted by their eigenvalues exceeding 2.5.

The principal component F1 correlated strongest with the occurrence of BbF, BkF, HCHO, DOC and BaP, which formed a uniform, closely correlated group. A slightly weaker connection existed between F1 and the occurrence of phenolic compounds and CHR. Component F2 was associated mainly with the presence of PHE, PYR, FL and FLA, and it was strongly and positively correlated with their concentrations. Similarly, other variables were also closely and positively correlated with one another, while they were only marginally correlated to the vectors representing F1 (Figure 4).

In Figure 4, there are two marked clusters of lakes characterised by similar features of the contaminants present in them. The first cluster, located in the third quarter of the coordinate system (where F1 and F2 are negative), was relatively uniform. It was characterised by a low concentration of most of the analysed compounds, while higher concentrations were only found for IcdP, BghiP and BaP (as compared to other lakes). To this cluster belonged the lakes located in the forefield of the Renard Glacier (R), in front of the Antonia Glacier (A_1, 2, 3) and some of those located in the Scott Glacier forefield (S_6) and in the valley of Logne Valley (L_1, 4, 5). The second cluster of investigated lakes (S) was not as compact and it was linked mainly to F2, in a rather wide range of concentrations of the chemical compounds representing this principal component. This cluster mostly included lakes from the forefield of the Scott Glacier (S), as well as single lakes from other areas (R_4, 13, 25, 26, 27 and L_2, 3). Furthermore, the PCA plot shows research objects that are very distant from the two clusters (S_3, S_13, R_1); these are characterised by much higher concentrations of particular contaminants than the other objects of this study (Table 2).

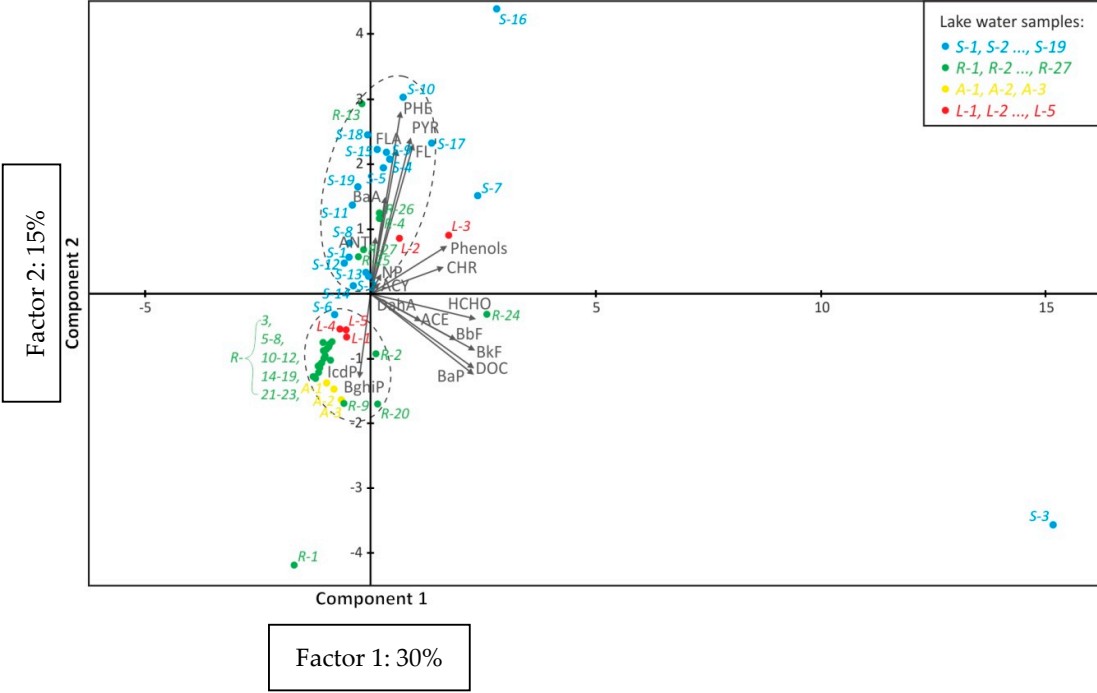

**Figure 4.** Variables and location of vectors in relation to principle components (F1 and F2). Lake water samples from Logne Valley (red dots), Scott Glacier (blue dots), Renard Glacier (green dots), Antonia Glacier (yellow dots).

### 3.3. PAH Indicator Ratio Analysis

In recent years there has been an increase in the number of studies concerning the presence and origin of organic compounds from the group of PAHs in Svalbard surface waters [33,35]. It is worth mentioning that PAH indicator ratios commonly used in the literature were used in this paper to assess the potential sources of PAHs in lakes located in NW Wedel Jarlberg Land (Table 3).

**Table 3.** PAH indicator ratio for surface water samples of 54 lakes collected in NW Wedel Jarlsberg Land.

| PAH Indicator Ratio | Potential Source of Pollution | Logne Lakes | Scott Lakes | Renard Lakes | Antonia Lakes |
|---|---|---|---|---|---|
| ANT/(ANT + PHE) | <0.1 Petrogenic | N = 3 | - | - | - |
| | >0.1 Pyrogenic | N = 2 | N = 1 | N = 1 | - |
| BaA/(BaA + CHR) | 0.2–0.35 Coal combustion | - | - | - | - |
| | <0.2 Petrogenic | - | - | - | - |
| | >0.35 Pyrogenic | N = 5 | N = 10 | N = 1 | N = 2 |
| FLA/(FLA + PYR) | <0.4 Petrogenic | N = 1 | N = 18 | N = 1 | - |
| | 0.4–0.5 Fossil fuel combustion | N = 2 | N = 1 | N = 7 | - |
| | >0.5 Grass, wood and coal combustion | N = 2 | - | N = 9 | N = 2 |
| IcdP/(IcdP + BghiP) | <0.2 Petrogenic | - | - | - | - |
| | 0.2–0.5 Petroleum combustion | - | - | N = 1 | - |
| | >0.5 Grass, wood and coal combustion | - | - | - | - |
| PHE/ANT | >10 petrogenic origin | N = 3 | N = 1 | - | - |
| | <10 pyrogenic combustion | N = 2 | - | N = 1 | - |
| FL/PYR | ≈1 Indicates pyrolytic origin | | N = 15 | N = 5 | - |
| | >1 Attributed to petrogenic source | N = 5 | - | N = 11 | - |
| BaP/BghiP | <0.6 Non traffic emissions | - | - | - | - |
| | >0.6 Traffic emissions | - | - | - | - |

| FL/(FL + PYR) | <0.5 Petrol emissions | - | N = 15 | N = 5 | - |
| | >0.5 Diesel emissions | N = 5 | - | N = 13 | N = 1 |

N—number of samples for which calculated values of PAH ratios correspond to a PAH indicator related to a specific source of pollutants.

PAH indicator ratios were presented for selected samples where the presence of compounds allows for calculation. In the lakes from Logne Valley and marginal zones of the Scott and Renard glaciers, results indicate the high impact of petrogenic sources of PAH compounds (67%, 68%, 72% of the received results, respectively) which could be related to human activity and ongoing degradation of the Arctic environment [4]. Interestingly, the indicator ratio (FLA/(FLA + PYR)) for the Logne and Renard lakes indicated a higher contribution of PAHs related to fossil fuel, grass, wood and coal combustion, while for the Scott lakes it indicated strictly petrogenic sources of PAHs. The situation was similar with the indicators (FL/PYR) and (FL/FL + PYR) which in the case of Logne and Renard suggested more petrogenic sources of compounds and their relation to diesel emissions, while for Scott lakes they indicate the pyrolytic origin of studied compounds and the contribution of petrol emission in sources of PAHs.

## 4. Discussion

### 4.1. PAH Compounds

In recent years, the number of studies proving that the Arctic is no longer a pristine environment free from anthropogenic pollution has increased [2,5,6]. Wet and dry deposition of atmospheric pollutants coming from both local sources and from transboundary transport from Eurasia and North America, is one of the main factors shaping the hydrochemistry of Arctic surface water [15,43,44,47]. PAHs, which can originate from natural and anthropogenic sources, reach Svalbard Archipelago mostly due to their transport in the atmosphere [33,48]. Usually, natural sources of PAH emission (e.g., volcanic eruptions, forest fires) are treated as insignificant background [49], but in cases of large-scale forest fires and intensive volcano activity natural emission cannot be ignored [4].

The results of PAH concentrations in Svalbard lakes were compared to those from other surface water collected in Hornsund Fjord in order to understand the magnitude of contamination. Based on the results of organic compounds presented in Table 2 and Figure 4, it may be noted that lakes from Logne Valley and marginal zones of the Scott and Renard glaciers were more polluted with PAH compounds than surface water from the Fuglebekken catchment in the year 2012 ($\Sigma_{16}$PAHs was in the range 13.3–104 ng/L). However, it is also worth noting that they were much less polluted than water samples from the Fuglebekken catchment in the years 2010 and 2011 ($\Sigma_{16}$PAHs were in the range 695–6 797 ng/L, 101–3 477 ng/L, respectively), which pollution is considered to be related to episodes of volcanic eruptions [4]. The years 2009, 2012 and 2013 are considered by Kozak et al. [4] to be free from volcanic eruption activity. For samples from the Fuglebekken catchment in 2012, the PAH diagnostic ratios indicate a high contribution of pyrogenic sources, as well as grass, wood and coal combustion sources, and also the contribution of petrogenic and diesel emission sources, which is similar to the origin of PAH compounds in the Logne and Renard lakes presented in this study.

There are only a few local potential sources of PAH compounds in Svalbard, such as human settlements (e.g., Longyearbyen, Ny-Alesund, Barentsburg and Sveagruva). Longyearbyen is mostly influenced by tourism and research activities, while Ny-Alesund changed its role from coal mining to a research station. In Barentsburg and Sveagruva there are coal mines, of which the Sveagruva mine is the main one on Spitsbergen [2].

An important aspect in understanding the spatial distribution of pollutants is sampling site exposure to wind direction. Despite the fact that the Sveagruva coal mine is located at the end of the Bellsund Fjord coast, the shape of the fjord favours the winds blowing from the west [50], which excludes Sveagruva as a potential source of PAHs in NW Wedel Jarlsberg Land (which is located to the south-west of Sveagruva). As presented in Figure 2, the prevailing wind directions before and

during the sampling campaign were mainly NW, WNW, ENE and E. The aspect of the Logne Valley, where PAH concentrations are highest, seems to favour the high exposure of the lakes to winds flowing from NW and WNW directions and thus to deposition of pollutants coming from the Greenland Sea. Groups of lakes from the forefields of the Scott and Renard glaciers are well hidden from the influence of winds coming from NW and NNW but slightly exposed to winds coming from ENE. Meanwhile, the Antonia sampling site is not exposed to winds from NW, WNW, ENE and E. It is only slightly exposed to winds from the south. Thus the exposure of the sampling site definitely favours deposition of pollutants coming from LRTAP in Logne valley lakes.

Based on the analysis of the chemical results and morphological conditions of the studied area, it can be noted that with the decrease in glacier area in the vicinity of the sampling site, levels of PAH detected in lakes in the basin of the glacier increased. This partially explains why the values of PAHs were highest in L and lowest in A, as well as the differences in concentrations of selected pollutants between the S and R sampling sites.

*4.2. Phenolic Compounds, HCHO and DOC*

In Svalbard surface waters, both phenolic compounds and HCHO were determined in glaciated [44,51,52] and non-glaciated catchments [35,43]. These compounds are considered to at least partially originate from anthropogenic, due to their presence in rainwater samples collected in Hornsund [53]. In 2012 they were also detected in the Scott river water and their presence was associated with the occurrence of precipitation [44]. In this paper, Scott lakes are distinguished by the highest concentration of phenolic compounds (mean concentration 0.323 mg/L) and HCHO (mean concentration 0.11 mg/L) of all lakes in the study area. In other lakes, the mean concentration of $\Sigma$phenols ranges from 0.020 to 0.064 mg/L and HCHO from 0.01 to 0.08 mg/L. Based on the literature, it may be noted that phenols occur more often in waters under the influence of glaciers than in those under the influence of permafrost [43,44]. This could explain the higher concentration of these compounds in the lakes located in the intra-marginal (Scott and Renard) and extra-marginal (Antonia) zones of the glaciers, in comparison to the very low levels in lakes located further from the glacier in the Logne Valley.

There are few papers examining concentrations of DOC in Svalbard streams and rivers fed by thawing permafrost [43] or melting glaciers [44,54,55]. Both the highest DOC value and the lowest were determined in the lakes located on the forefield of the Scott Glacier. This corresponds to the PCA analysis results, where S results are the least uniform, while lakes from the R group are very uniform thanks to the similar values of organic compounds determined in these lakes. Mean levels of DOC in the group of lakes from the Logne Valley and the Antonia Glacier marginal zone are higher than those noted in the intra-marginal zones of the Scott and Renard glaciers. However, they do not correspond to mean values of the $\Sigma_{16}$PAHs detected in this study and do not show a relation with the concentration of phenolic compounds or HCHO in lakes. They also do not correspond with the possible influence of wind directions, nor with the morphometric parameters of the glaciers known to the authors, or to the location of sampling points. This indicates the influence of lake-specific environmental conditions unknown to the authors, and also the high presence of other organic compounds in the lakes' waters that might correspond with DOC.

## 5. Summary

This study shows that the concentrations of PAHs determined in lakes from NW Wedel Jarlsberg Land in the year 2012 are the result of such factors as: (1) exposure of sampling site to winds which could determine the transport and deposition of the transboundary pollutants from areas of lower latitudes; (2) levels of PAHs increasing with a decrease in the area of the glacier in the glacier basin, and being higher in samples located closer to the glacier. Moreover, most of the PAH indicator ratios suggest that the detected compounds are of pyrogenic origin related to human activity. Concentrations of phenolic compounds and HCHO mostly occur in lakes in the close vicinity of small glaciers such as the Scott Glacier.

**Author Contributions:** Conceptualization, S.L.-K. and Ż.P.; methodology, S.L.-K. and Ż.P.; validation, S.L.-K.; formal analysis, Ł.F.; investigation, S.L.-K.; resources, S.L.-K., Ż.P., and Ł.F.; data curation, S.L.-K.; writing—original draft preparation, S.L.-K., Ł.F. and M.R.; writing—review and editing, S.L.-K., Ł.F. and M.R.; visualization, S.L.-K. and Ł.F.; supervision, Ż.P.; project administration, S.L.-K. and Ż.P.; funding acquisition, S.L.-K., M.R., Ł.F. and Ż.P. All authors have read and agreed to the published version of the manuscript.

**Funding:** This research was funded by National Science Centre of Poland, grants number 2011/01/B/ST10/06996 and 2015/17/N/ST10/03177, and by Ministry of Science and Higher Education of Poland, grant number N N 306 703840.

**Acknowledgments:** Special thanks to Grzegorz Gajek and Leszek Łęczyński for their tremendous help in the fieldwork and collection of samples.

**Conflicts of Interest:** The authors declare no conflict of interest.

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
