# Peer review of "Contamination of Arctic Lakes with Persistent Toxic PAH Substances in the NW Part of Wedel Jarlsberg Land (Bellsund, Svalbard)"

_water, doi:10.3390/w12020411_

Round 1
Reviewer 1 Report
I have two types of technical comments concerning the paper (1) presentation and (2) significance
Presentation: there are a few areas where the graphic used are difficult to read and hinder comprehension of the material:
(1) place name labels on fig 1 are not in English, forcing the reader to guess at the correspondence with English place names given in the text. This is not impossible, but makes the reader work harder than necessary, especially given that location and orientation play a big role in the paper’s conclusions
(2) lat and lon values are given in the text but not shown on fig 1, again making it difficult to go back-and-forth between text and the figure
(3) in the discussion section, some of the chemical variations are linked to relative distance of some of the lakes to the glacier front. This is hard for the reader to evaluate because lakes are almost impossible to see on fig. 1. There are red boxes on fig 1 that suggest that perhaps there might have been a second, more detailed index map of the 4 study areas created but never used? At the vey least, the authors could consider symbolizing some of the lake sampling locations on fig 1
(4) Table 3 - Perhaps it is just the way the authors have formatted this table on the page, but I find this table difficult to read and difficult to make comparisons between the sites. It has to do with how some entries wrap to a second line. Also on Table 3 – I assume numbers in parentheses are citations to data collected by others and published in the cited works? If so, this is certainly not made clear in the text – that some of the “results” are from other studies. Maybe I missed something here.
(5) Figure 3: I recognize the challenge of presenting and graphically symbolizing results from more than a dozen components, but I think the author’s choice of the 3D column diagram actually hinders understanding. The inconsistent colors on the cylinders that result from shading and the bevel on the explanation color boxes make it difficult to pick out which chemical species are being plotted and compared. The authors could consider labeling some of the species on the columns themselves and should consider condensing this plot to a more traditional 2D plot.
(6) The authors could do a better job of helping the reader understand the principal components analysis plot (fig 4). It is only by guessing that the reader realizes that the colored dots are samples – this should be explained. It seems to me that there are sample groupings mentioned in the text (for instance the Scott Glacier samples such as S_3, S_13) that are not shown on the plot or at least not shown on the explanation.
(7) Table 4 exactly repeats the indicator ratio criteria that were presented in Table 2. The authors should consider deleting Table 2, which would give them some room to add a little text. I would appreciate a few sentences reviewing the published derivation of these ratios and how they came to define indications of specific contaminant sources.
Significance:
(1) It is not clear to me that this study does more than confirm previous work. The authors point out in the introduction that there are at least two Svalbard-specific publications on pretty much every chemical grouping that they analyzed. So what is new? Were there deficiencies in the previous work? It is not clear to me what “gap in knowledge” this paper is filling
(2) I am not convinced that valley orientation and measurement of local wind direction is sufficient to interpret the differences in concentrations of organic compounds in the four valleys. I accept that the compounds are likely sourced from lower latitudes and that the indicator ratio criteria provide clues about the types of processes that released the compounds. However, understanding the spatial variability in contaminant composition between these valleys would require (a) a rigorous accounting of the types of contaminant sources, both on the island and farther afield, (b) an understanding of the weather patterns in the region surrounding Svalbard during the field season, and (c) an understanding of the atmospheric circulation and degree of mixing in the atmosphere over a broad region including Svalbard. I view the measured wind direction at the mouth of the valley as a local phenomenon; I would assume that the air masses approaching Svalbard from low latitudes would be pretty effectively mixed. So what would cause the local variability? It would be surprising if contaminants from low latitudes could remain unmixed in their journey to Svalbard such that different valleys had different composition.
Reviewer 2 Report
Dear Authors,
I found your work titled "Contamination of Artic lakes....." very interesting and, due to its scientific content, worth of publication after some minor changes.
However, the English is definitely below the standard required for the publication on an international journal. A lot of sentences are obscure, affected by missing or erroneous punctuations, and these flaws do not give the right merit to the scientific content of your work.
I warmly suggest a deep linguistic revision by a professional proofreading service or an English mother tongue person.
Please find attached a commented pdf with some comments maybe useful for improving your manuscript
